# Genome-wide gene expression noise in *Escherichia coli* is condition-dependent and determined by propagation of noise through the regulatory network

Arantxa Urchueguía[1,2], Luca Galbusera[1,2], Dany Chauvin[1,2], Gwendoline Bellement[1,2], Thomas Julou[1,2]*, Erik van Nimwegen[1,2]*

1 Biozentrum, University of Basel, Basel, Switzerland, 2 Swiss Institute of Bioinformatics, Basel, Switzerland

* thomas.julou@unibas.ch (TJ); erik.vannimwegen@unibas.ch (EvN)

**Data Availability Statement:** All the raw and processed data used in this study, including metadata describing the experiments, and

## Abstract

Although it is well appreciated that gene expression is inherently noisy and that transcriptional noise is encoded in a promoter's sequence, little is known about the extent to which noise levels of individual promoters vary across growth conditions. Using flow cytometry, we here quantify transcriptional noise in *Escherichia coli* genome-wide across 8 growth conditions and find that noise levels systematically decrease with growth rate, with a condition-dependent lower bound on noise. Whereas constitutive promoters consistently exhibit low noise in all conditions, regulated promoters are both more noisy on average and more variable in noise across conditions. Moreover, individual promoters show highly distinct variation in noise across conditions. We show that a simple model of noise propagation from regulators to their targets can explain a significant fraction of the variation in relative noise levels and identifies TFs that most contribute to both condition-specific and condition-independent noise propagation. In addition, analysis of the genome-wide correlation structure of various gene properties shows that gene regulation, expression noise, and noise plasticity are all positively correlated genome-wide and vary independently of variations in absolute expression, codon bias, and evolutionary rate. Together, our results show that while absolute expression noise tends to decrease with growth rate, relative noise levels of genes are highly condition-dependent and determined by the propagation of noise through the gene regulatory network.

## Introduction

It is by now well established that isogenic cells growing in a homogeneous environment show cell-to-cell fluctuations in gene expression (for example, [1–4]). This gene expression noise is not surprising from a biophysical perspective, given the inherent thermodynamic fluctuations in the molecular events underlying gene expression and the small numbers of molecules involved. In the simplest models of gene expression, where promoters are transcribed at a constant rate, the "intrinsic" noise in gene expression would simply grow in proportion to the

annotation data, are freely available from https://doi.org/10.5281/zenodo.5701705.

**Funding:** This work was partly funded by the Werner Siemens Stiftung through a fellowhip to AU, the Swiss National Science Foundation SystemsX.ch StoNets grant, and the Swiss National Science Foundation grant 31003A_159673 to EvN. The funders had no role in study design, data collection and analysis, decision to publish, or preparation of the manuscript.

**Competing interests:** The authors have declared that no competing interests exist.

square root of a gene's absolute expression level (for example, [5]). However, even in bacteria where the gene expression process is considerably simpler than in eukaryotes, genes typically exhibit significantly higher levels of transcriptional noise, indicating that transcription rates fluctuate in time and across cells due to "extrinsic" factors [1]. Moreover, studies of genome-wide gene expression noise in bacteria have shown that genes with the same absolute expression can exhibit different noise levels and that the transcriptional noise of a gene is to a substantial extent encoded in its promoter sequence [6–9]. However, how the promoter sequence of a gene determines its transcriptional noise and what factors are the main drivers of differences in transcriptional noise remains largely unknown.

In addition, because genome-wide studies have so far focused on gene expression noise in a single growth condition, it is currently not clear to what extent gene expression noise in bacteria is condition-dependent. That is, we do not know to what extent absolute noise levels vary across growth conditions and whether genes with the highest noise in one condition also exhibit the highest noise in other conditions.

A systematic investigation into the condition dependence of genome-wide gene expression noise may provide important insights into what drives both absolute and relative noise levels of promoters. For example, it is possible that transcriptional noise is mostly driven by fluctuations in general factors, for example, the concentrations of RNA polymerases and nucleotides, and the overall state of the DNA. For example, it has been suggested that noise levels in yeast are mainly determined by basic promoter architecture and associated nucleosome positioning (see [10] and citations therein). Similarly, since supercoiling of the DNA has been reported to control the sizes of transcriptional bursts in *Escherichia coli* [11], it is conceivable that a promoter's noise properties depend on its sensitivity to supercoiling. If differences in transcriptional noise across promoters result mainly from differences in the sensitivity of promoters to such global factors, then one would expect the same promoters to show highest noise across conditions.

Alternatively, instead of a promoter's noise level being an intrinsic feature of its architecture, a promoter's noise might be determined by the way it is regulated in a given condition. Since the transcription rate of a promoter will generally depend on the binding of transcription factors (TFs), a promoter's transcription rate will fluctuate as TFs stochastically bind and unbind to it. The rates of binding and unbinding of TFs in turn depend on average expression levels and fluctuations in expression levels of TFs across cells [8,12–14]. Consequently, fluctuations in both the expression levels of TFs and their binding to promoter regions will thus unavoidably propagate to fluctuations in expression of their target genes [15–19].

That noise propagation may play an important role for genome-wide gene expression noise was suggested by results we obtained in a previous study in which we measured genome-wide gene expression noise of *E. coli* promoters in a single growth condition and compared this with expression noise of synthetic promoters that were selected from a large library of 100 to 150 bp random sequence fragments [9]. We not only found that the synthetic promoters generally exhibited low expression noise, but also found that native promoters with high expression noise tended to have more known regulatory inputs from TFs than genes with low expression noise. To explain these observations, we developed an evolutionary theory in [9] explaining why natural selection may favor noisy gene regulation in many situations. However, to what extent genome-wide gene expression noise is indeed determined by noise propagation is currently unclear, and one of the motivations of this study is to systematically investigate this experimentally.

As TFs change their expression levels across growth conditions, so will the fluctuations in their binding at their target promoters. Consequently, a key characteristic that distinguishes noise propagation from other sources of expression noise is that this noise will be highly

condition-dependent. Therefore, a systematic investigation of how genome-wide noise levels of promoters vary across condition should directly provide insights into the role of noise propagation.

To investigate the condition dependence of gene expression noise and elucidate the roles of both global factors and noise propagation, we systematically quantified genome-wide gene expression noise in *E. coli* across 8 different conditions that represent a wide range of growth rates and include different nutrients, different types of stress, and stationary phase.

## Results

### Expression noise levels vary substantially across conditions and systematically decrease with growth rate

Using methodology already employed in several previous studies [7,9,20], we used flow cytometry together with a library of fluorescent transcriptional reporters [21] to measure gene expression distributions of *E. coli* promoters genome-wide across a set of 8 different growth conditions (Fig 1A). The library of fluorescent reporters consists of most of *E. coli*'s intergenic regions inserted upstream of a strong ribosomal binding site and a fast-folding GFP on a low copy number plasmid. As we have shown previously [9], the GFP levels of these reporters reflect transcriptional activity, since translation and mRNA decay rates vary little across these reporters, which have almost identical mRNAs.

The growth conditions (see SI Methods and Texts in S1 Text) were chosen to span a wide range of growth rates (Fig A in S1 Text), cell physiologies (Fig B in S1 Text), and regulatory states. They consist of MOPS synthetic rich media, M9 minimal media with 3 different carbon sources (0.2% glucose, 0.2% glycerol, and 0.2% lactose), 2 stresses (sub-MIC antibiotic: ciprofloxacin 1.5 ng/ml + 0.2% glucose and osmotic: 0.4 M NaCl + 0.2% glucose), and 2 time points in stationary phase (after 16 h and 30 h of growth in 0.2% glucose, respectively). We used microscopy to image cells from each growth condition and found that, consistent with the known relationship between growth rate and cell physiology [22], cell size generally increased with growth rate (Fig C in S1 Text).

For each condition and each promoter, we used high-throughput flow cytometry to measure GFP levels for thousands of single cells. Apart from the 2 stationary phase conditions, all measurements were taken during mid-exponential phase. In total, we gathered 50′000 single-cell measurements for each of the 1,810 promoters in the library across 8 conditions, including some conditions in replicate. As observed previously [9], the fluorescence distributions can be well fitted with log-normal distributions, and we thus characterized each fluorescence distribution by the mean and variance of log-fluorescence. We note that, since flow cytometry measurements are themselves noisy, inferring means and variances from the raw measurements requires careful computational procedures, and we here use a set of procedures that we recently developed [23]. These include using forward and side scatter to identify events corresponding to cells and fits the log-fluorescence distribution by a mixture of a Gaussian and uniform distributions to remove possible outliers (for example, contaminants and nongrowing cells), as described in [23].

Replicate measurements performed on different days were highly reproducible, with Pearson squared correlations $R^2 > 0.99$ for the mean between replicates in all conditions and squared correlations for the variance ranging from $R^2 = 0.85$ to $R^2 = 0.95$ (Fig D in S1 Text). In order to determine whether this variability derived mainly from biological variation from day to day or from measurement noise, we performed a time course experiment where we repeatedly measured the same culture at different time points during exponential growth and found that both the mean and variance measurements were extremely reproducible in these

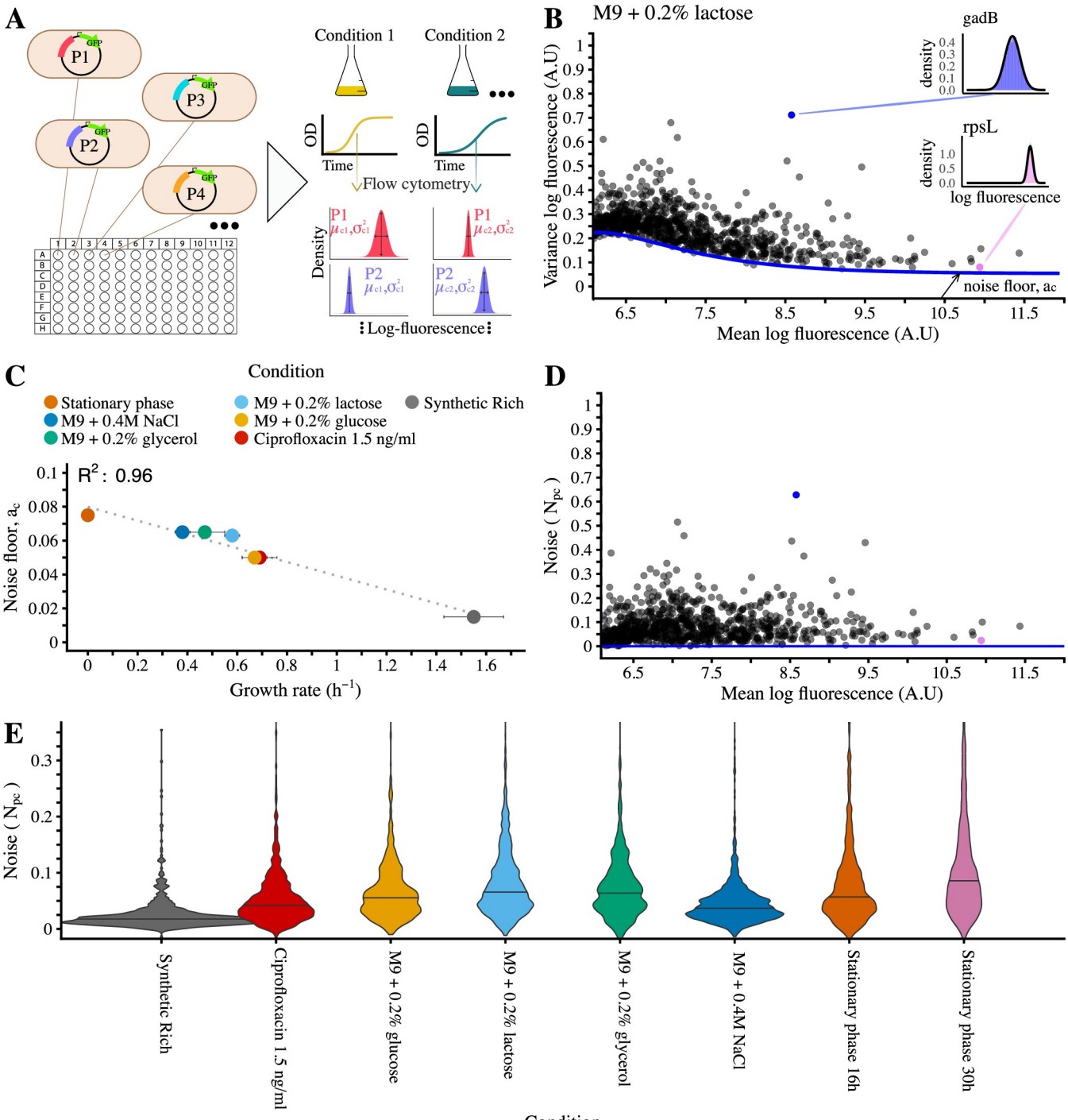

**Fig 1. Genome-wide expression noise of *E. coli* promoters varies significantly with growth condition.** (A) For each growth condition and *E. coli* promoter, we used flow cytometry to measure the distribution of GFP levels across single cells of the corresponding fluorescent reporter. The 8 growth conditions comprised synthetic rich media, minimal media with different carbon sources, an osmotic and DNA damage stress, and 2 time points in stationary phase. (B) Mean (x-axis) and variance (y-axis) of log GFP levels for all promoters with expression above background level for growth in M9 0.2% lactose (see Fig G in S1 Text for results in all conditions). The blue line shows the fitted minimal variance as a function of mean expression and the corresponding noise floor $a_c$ is indicated with an arrow. The insets show distributions of log-GFP levels for 2 example promoters. (C) The noise floor $a_c$ as a function of the growth rate in the respective condition (stationary phase at 30 h not shown). The dotted line indicates a linear fit (with Pearson squared correlation coefficient $R^2$ indicated). (D) To compare noise of promoters with different means, we defined the noise level of a promoter as the difference between its variance and the fitted minimal variance at its mean expression. Shown are noise levels versus mean for promoters in M9 0.2% lactose. (E) Noise level distributions of the full library in each of the measured conditions. The horizontal lines indicate the medians. The vertical scale is clipped at 0.35 for better visibility (Fig H in S1 Text has the full distributions). The underlying data for Fig 1 can be found in S1 Data and https://doi.org/10.5281/zenodo.4662163.

experiments (Fig E and F in S1 Text). This implies that variation in measurements from different days are mostly due to uncontrolled biological variation, and not measurement noise. This also implies that genes exhibit more biological variation in their noise levels across days than in their mean expression.

To illustrate the typical form that the distribution of means and variances of log-expression across promoters takes, Fig 1B shows the variance as a function of mean for each promoter measured in M9 minimal media + 0.2% lactose (see Fig G in S1 Text for all conditions). Note that the variance in log-expression is equal to the square of the coefficient of variation ($CV^2$) whenever fluctuations are small relative to the mean [9]. This approximation applies in our data, as the majority of promoters (approximately 75% across all conditions) have a variance smaller than 0.3 (Fig G in S1 Text).

As has been observed in previous studies [6,9,24,25], we find that there is a clear lower bound on noise as a function of the mean expression level of the promoter (Fig 1B), which decreases with mean, and asymptotes to a fixed lower bound at high mean expression. A qualitatively similar curve is observed in all growth conditions (Fig G in S1 Text). As derived previously [9] and explained in the S1 Text, the functional form of the minimal variance as a function of mean expression can be derived, assuming that GFP variance is the sum of 2 terms: one "multiplicative" contribution with variance proportional to the square of the mean expression, and one "Poissonian" contribution with variance proportional to mean expression. The Poissonian term, whose magnitude we denote by $b_c$ and is often referred to as the "intrinsic noise" term, could in principle derive from intrinsic expression noise whose magnitude scales proportional to mean expression [6,26].

However, by comparing microscopy and flow cytometry measurements we have recently shown that, at these expression levels, the component $b_c$ derives almost entirely from the measurement noise of the flow cytometer [23]. We will refer to the multiplicative term as the "noise floor" $a_c$, which is often referred to as an "extrinsic noise" contribution. In contrast to the Poissonian term, whose contribution decreases with increasing mean and is negligible for highly expressed promoters, the contribution of the noise floor is independent of expression mean and corresponds to the minimal variance for highly expressed promoters. As shown in Fig G in S1 Text, the same functional form describes the minimal variance in all conditions, and we estimated the noise floor $a_c$ for each condition.

We observed that the noise floor $a_c$ systematically decreases with growth rate over the entire range of growth rates. Although we currently lack a theoretical model for how this noise floor depends on growth rate, we noted that the dependence is well fit by a simple linearly decreasing function ($R^2$ = 0.96; Fig 1C). However, we stress that this is only a phenomenological observation valid for the growth conditions considered here and that it is currently unclear whether this relationship generalizes to other conditions, for example, when growth rate is modulated by subinhibitory levels of antibiotics. The noise floor $a_c$ likely reflects the minimal noise that every promoter is subject to due to general fluctuations in the physiological state of the cell including overall transcription, translation, mRNA decay, and growth [1,6]. Since we are measuring total protein levels per cell, one possible contribution to the noise floor is the variation in cell sizes. Although average cell size increases systematically with growth rate (Fig C in S1 Text), we find that the coefficient of variation of cell size does not vary much across conditions and shows no correlation with either the growth rate or the noise floor (Fig I in S1 Text). Therefore, changes in the cell size distribution do not explain the decrease of the noise floor with growth rate.

Since our reporter constructs use a low copy number plasmid, some of the observed variation in expression levels may derive from plasmid copy number fluctuations. We note that, since the only differences between the reporter constructs are the short promoter sequences

upstream of the GFP gene, all differences in the log-expression means and variances of different promoters within a given condition must be due to the differences in their promoter sequences. However, it is conceivable that plasmid copy number variations contribute significantly to the noise floor across conditions. As detailed in the S1 Text, we tested this hypothesis by selecting a set of promoters that were observed to have noise near the noise floor across all conditions, created chromosomal constructs for these promoters, and systematically compared mean and variance in log-expression of these chromosomal constructs with the corresponding plasmid-based reporters. As shown in Fig J in S1 Text, we find that whereas mean expression levels of the plasmid reporters are consistently about 6.5 times higher than the corresponding chromosomal reporters, the noise levels of the plasmid and chromosomal constructs are very similar, with differences generally within the error bars. These results show that plasmid copy number noise is either similar to the chromosomal copy number noise or that the copy number noise is small compared to other factors that determine the noise floor.

An anticorrelation between noise and growth rate, similar to the one we observe here, has previously been observed in eukaryotes but was proposed to derive from heterogeneity in cell cycle stage [27]. However, our results show that this general anticorrelation between noise and growth rate also occurs in prokaryotes that do not have analogous cell cycle stages.

In order to have a measure of the relative levels of noise of genes that is not confounded by the systematic dependence on mean expression, we defined the noise level $N_{pc}$ of promoter $p$ in condition $c$ as the difference between its variance in log-fluorescence and the noise floor, that is, the minimal variance at its mean expression level (see S1 Text, equation (3)). As shown in Fig 1D, the noise levels $N_{pc}$ indeed no longer show any systematic dependence on mean expression, and this is observed across all conditions (Fig G in S1 Text).

Fig 1E shows the distribution of noise levels $N_{pc}$ in each of the conditions, sorted from high to low growth rate. We see that not only the noise floor, but also the distribution of noise levels on top of this noise floor varies substantially across conditions. Moreover, like the noise floor, both the median of the noise levels $N_{pc}$ as well as the variability in noise levels increase as the growth rate decreases, for example, the noise levels are lowest in synthetic rich conditions ($p = 3 \times 10^{-30}$, Wilcoxon rank-sum test) and highest at 30 h of stationary phase ($p = 5 \times 10^{-68}$, Wilcoxon rank sum test). That is, not only do minimal noise levels increase as growth rate decreases, the variability in noise levels across genes increases as well. The only exception to this general trend is the osmotic stress condition M9 + 0.4 M NaCl, which has relatively low variability in noise levels $N_{pc}$ compared to other conditions with similar growth rate (Fig 1E), even though its noise floor is not deviating from the general dependence on growth rate.

These results show that the physiological state of the cell has a major influence on the distribution of absolute noise levels and that both the mean and variation in noise levels generally decreases with growth rate. We now turn to investigating how the relative noise levels of different promoters vary across the measured conditions.

## Individual promoters show highly diverse changes in noise across conditions

If changes in noise levels across conditions were mostly driven by fluctuations in global factors such as concentrations of RNA polymerase, we would expect different genes to exhibit coherent changes in noise across conditions. For example, relative noise levels of different genes may remain relatively unchanged across conditions, or alternatively, noise levels might rescale across conditions as a function of the mean expression of the gene in the condition. However, this is not what we observe. Instead, different promoters show highly diverse changes in their noise levels across conditions (Fig 2).

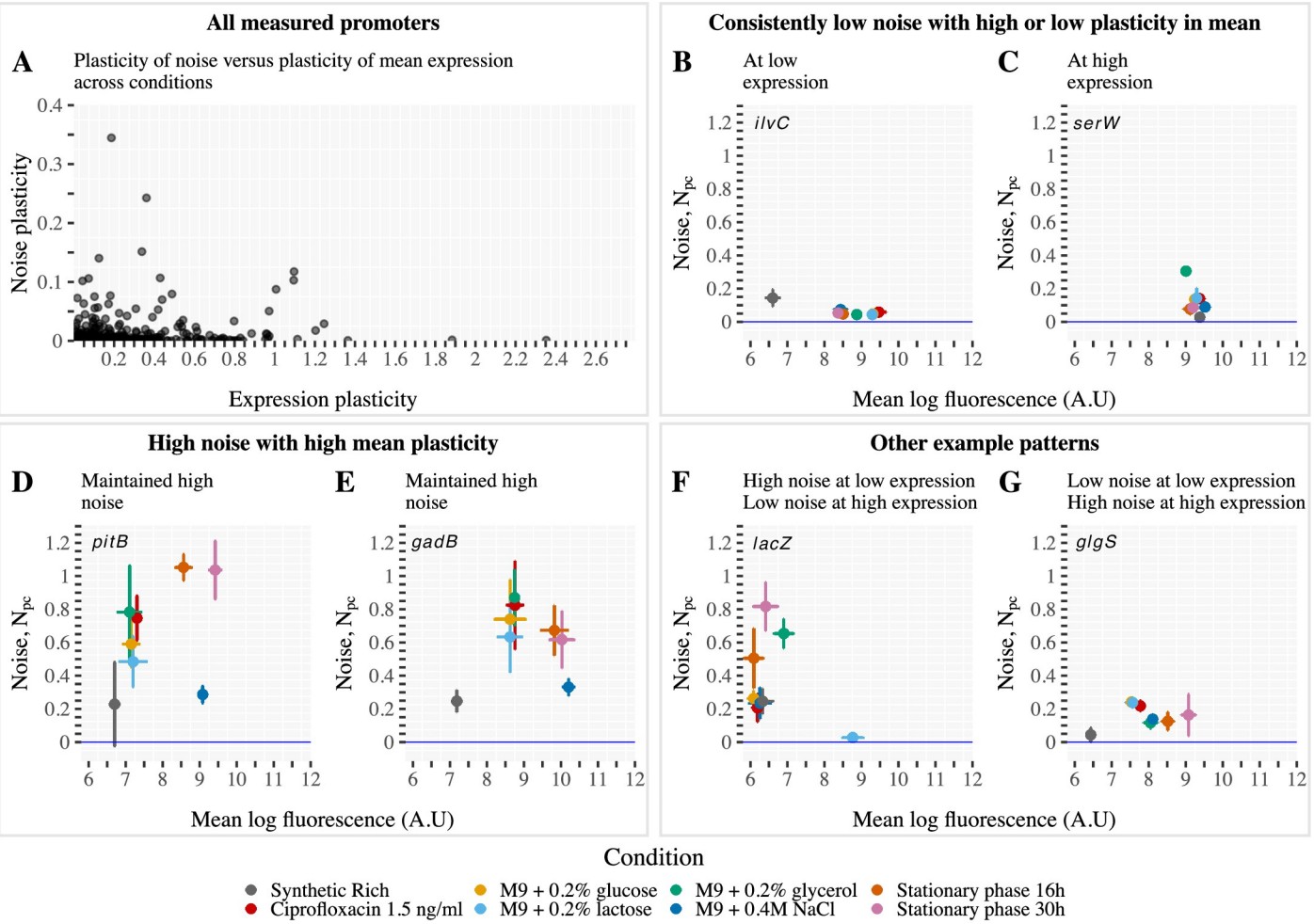

**Fig 2. Individual promoters show diverse patterns of variation in noise levels across conditions. (A)** Scatter plot showing the expression plasticity (variance across conditions, horizontal axis) and noise (variance in noise across conditions) of all measured promoters. **(B-G)** Examples of condition-dependent mean and noise of individual promoters. Each panel shows the noise level as a function of mean across conditions (colors; see legend) for one promoter, with the gene regulated by the promoter indicated in each panel. Error bars denote standard errors of the estimates based on biological replicate measurements. Each of the 3 pairs of panels indicate different types of behavior in mean and noise across conditions, as described at the top of each pair of panels. The underlying data for Fig 2 can be found in S1 Data and https://doi.org/10.5281/zenodo.4662163.

Following the general usage of the word plasticity to refer to the adaptability of the phenotype to changes in the environment, we will refer to the variance of a promoter's mean and noise level across conditions as the plasticity of its mean and noise. The plasticity of both mean and noise vary over a substantial range across promoters, without any clear systematic dependence between these quantities. Analogous scatter plots for the variation and dependence between average expression, average noise, and the plasticities in mean and noise show that all these quantities vary substantially across promoters (Fig K in S1 Text). That is, individual promoters show highly distinct variation in their mean and noise across conditions, and Fig 2B–2G shows some examples of the different behaviors we observe. Note that all observations in these panels have error bars that show the standard error of measured mean and noise across biological replicates. We observe promoters that are low noise in almost all conditions, either with high plasticity in mean (Fig 2B) or low plasticity in mean (Fig 2C). Other promoters show high noise with plasticity in both the mean and noise level, without clear correlation between

mean and noise level (Fig 2D and 2E). But many other patterns of behavior can be observed, such as promoters that show only low noise when the promoter has high mean (Fig 2F) or only low noise when the promoter has low mean (Fig 2G).

The growth media were not predictive for how individual genes were going to change their mean and noise. For example, while overall the whole library is shifted towards lower noise in synthetic rich media, individual genes can show higher noise in this condition compared to other conditions (for example, Fig 2B and 2F). We highlighted this particular condition as an example, but the same observation applies to others. These observations indicate that global changes in the cell physiology or in the expression level cannot explain how the noise of a promoter varies across conditions. This implies that there is a promoter-specific source of noise shaping condition-dependent gene expression variability. Just as the plasticity in mean expression derives from gene regulation, one obvious hypothesis is that this promoter-dependent source of condition-dependent noise derives from gene regulation as well.

## Noise propagation predicts that relative noise levels are condition-dependent

As mentioned in the introduction, the mechanistic basis for gene expression regulation is that the binding and unbinding of TFs to a promoter causes the transcription rate from this promoter to change. Consequently, fluctuations in the expression levels of TFs and their binding to promoter regions will unavoidably propagate to fluctuations in the expression of their target genes [8,12–19]. While the general decrease of absolute noise levels with growth rate (Fig 1C and 1E) is likely due to general physiological fluctuations that affect all promoters, the highly diverse changes in the relative noise levels of different promoters across conditions (Fig 2) is exactly what is expected to occur under a noise propagation scenario (Fig 3).

Let us consider a simple scenario in which 2 individual genes are each regulated by one TF, that is, gene A is regulated by TF1 and gene B by TF2 (Fig 3A). As the activities of these TFs fluctuate within a given condition, these fluctuations can propagate to their respective targets. For example, in a condition where TF1 exhibits less variation in activity from cell to cell than TF2, gene A will generally exhibit less expression noise than gene B (Fig 3A). In anticipation of analysis presented below, it is important to stress that the distribution of "TF activity" shown in Fig 3A is only a schematic representation of a much more complicated biophysical process at the molecular level, and different target promoters of the same TF might respond very differently to fluctuations in the TF's "activity." Roughly speaking, the extent to which a TF X will propagate noise to a given target promoter Y depends on how much the binding of TF X to promoter Y fluctuates in time and across cells and how much the transcription rate of promoter Y depends on these fluctuations in binding of TF X. For example, if promoter Y is already strongly repressed or activated by another TF, the binding of TF X may be irrelevant for its transcription, and TF X will not propagate noise to promoter Y. Even if the transcription rate of Y is sensitive to binding of TF X, it may still be that binding affinities of the sites in promoter Y are so weak that the promoter is essentially never bound or so strong that it is essentially always bound, even if the concentration of TF X fluctuates from cell to cell. Only those target promoters of X for which the transcription rate is both sensitive to the binding of TF X, and for which the binding of TF X fluctuates significantly, will experience significant increase in their noise levels. Thus, the amount of noise propagation from a given TF X to a given target promoter Y is a complex context-dependent function, and only a subset of the promoters that are targeted by TF X will indeed respond to fluctuations in the activity of TF X in a given condition.

These considerations make clear that, in general, we expect the extent to which different TFs propagate noise to different target promoters to be highly condition-dependent. For

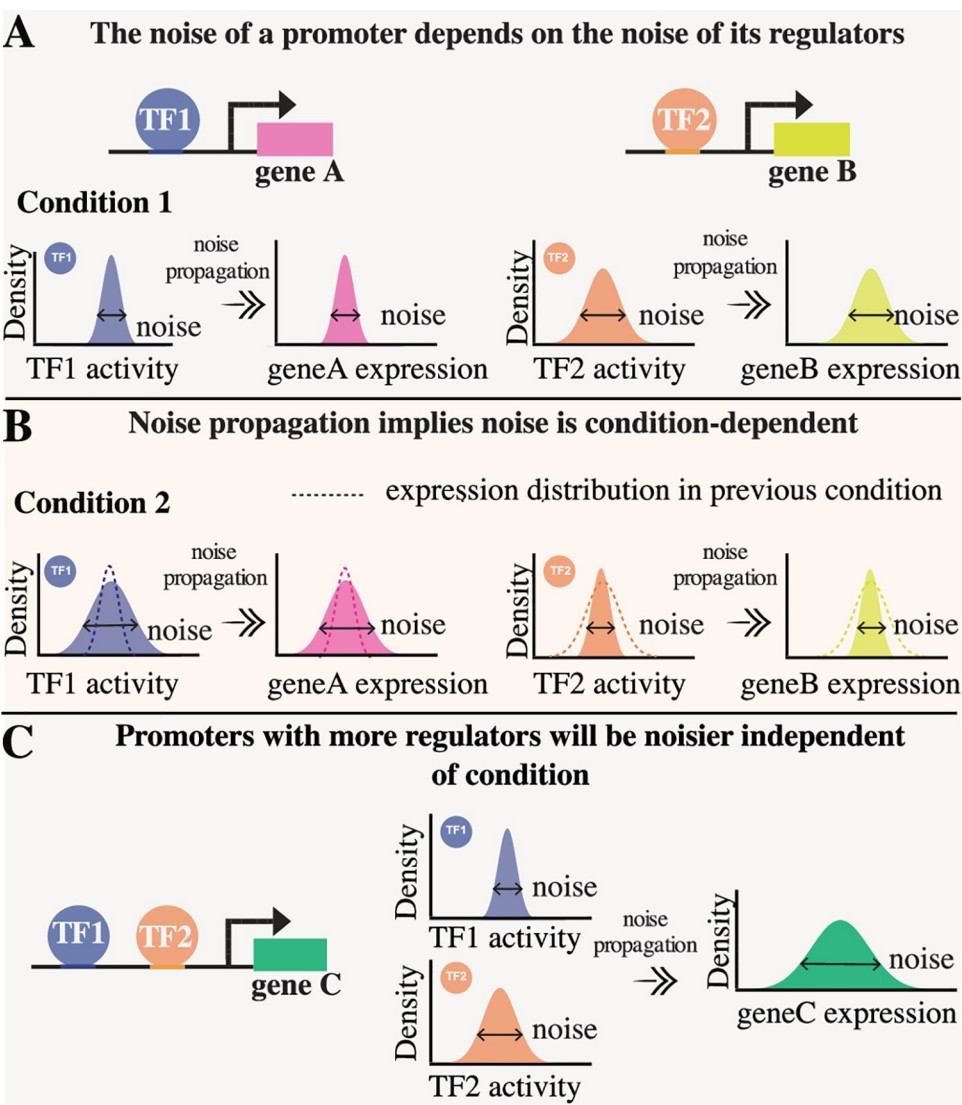

**Fig 3. Signatures of condition-dependent noise propagation. (A)** We imagine a scenario in which 2 promoters are each regulated by a single transcription factor (TF1 or TF2). In growth condition 1, TF2 shows a higher variability in its activity (orange distribution) than TF1 (blue distribution). As a result, its target (gene B, yellow) will show higher expression variability than the target of TF1 (gene A, pink). **(B)** If the relative levels of variability in the activities are reversed in a different condition, the relative noise levels of target genes A and B will likewise be reversed. That is, noise propagation can explain why transcriptional noise is highly condition-dependent. **(C)** Because the noise of a target gene depends on fluctuations in activities of all of the TFs that regulate it, promoters that are more regulated will typically show higher noise levels in all conditions. The illustration shows a promoter controlling the expression of gene C (green) which is regulated both by TF1 (blue) and TF2 (orange). Since at least one of these TFs is highly variable in each condition, gene C will exhibit high noise levels in both conditions.

example, for the simple scenario imagined in Fig 3A, we can easily imagine that, in another condition, TF1 may show higher variability than TF2, such that the noise levels of their targets would change accordingly (Fig 3B). In other words, if gene expression noise is to a large extent determined by noise propagation from regulators to their targets, then this would explain why relative noise levels of genes can vary in a complex manner across conditions, because we expect both the noise levels of different regulators and the sensitivity to this noise at different promoters to vary across conditions. In summary, we propose that the qualitative patterns in

expression noise across conditions that we observed in Fig 2 and Fig K in S1 Text can be explained by assuming that noise levels are to a large extent determined by propagation of noise from regulators to their targets.

The hypothesis that noise propagation is responsible for the observed condition-dependent relative noise levels makes a number of additional predictions. First, constitutive promoters, that is, promoters that are not targeted by any TF other than the sigma factor of the RNA polymerase, should exhibit low noise in each condition and relatively little plasticity in their noise levels. Second, the larger the number of regulators that target a given promoter, the larger the chance that the promoter will be sensitive to fluctuations in the activities of at least one of these TFs (Fig 3C). Thus, more noisy promoters are in general expected to have more regulatory inputs. In addition, because all regulatory inputs of a promoter can change their noise levels in a condition-dependent manner, we also expect that, the more regulatory inputs a promoter has, the higher the plasticity of its noise level will be. Finally, to the extent that the regulatory inputs of each promoter are known, it should be possible to explain why some promoters are more noisy in one condition, and other promoters more noisy in another condition, and identify which TFs are most responsible for noise propagation in different conditions. In the next section, we investigate whether our data indeed exhibit these properties.

## Noise propagation explains the condition-dependent noise levels of genes

In a previous work [9], we found that, for cells growing in minimal media with glucose, more noisy genes generally have more regulatory inputs, and we here checked whether these observations generalize to multiple growth conditions. We sorted promoters by their noise levels and used the regulatory site annotation from RegulonDB [28] to calculate the average number of known regulatory inputs of genes with noise levels $N_{pc}$ above a certain cutoff level, as a function of the cutoff level (Materials and methods). We find that in all 8 conditions, the number of known regulatory inputs systematically increases with noise levels (Fig 4A and Fig L in S1 Text). Notably, these differences are highly statistically significant with *t*-statistics of 4 or higher for the difference between known regulatory inputs for promoters above and below a given noise cutoff across a wide range of cutoffs in each condition (Fig M in S1 Text).

Next, we wanted to test whether constitutive promoters exhibit consistently low noise levels. This analysis is complicated by the fact that our knowledge of *E. coli*'s regulatory network is extremely incomplete, with no known target promoters for almost two-thirds of *E. coli*'s TFs. Thus, although no known regulatory input is known for almost 60% of *E. coli* promoters (Fig N in S1 Text), a substantial fraction of these promoters are likely regulated by TFs for which we currently lack information. To obtain a set of promoters that are very likely constitutive we took a random selection of synthetic promoters that we obtained previously by screening a library of 100 to 150 bp random sequence fragments for sequences that drive expression in M9 minimal media with glucose [9] (see Supplementary Methods in S1 Text). We measured mean expression and expression noise of these synthetic promoters across 4 growth conditions and compared their expression plasticity, average noise, and noise plasticity with those of native promoters that have at least one known regulatory input. We found that the synthetic promoters not only have lower expression plasticity (*p*-value = 1.545e-09, two-sided Welch's *t* test), confirming that they are likely constitutive but that both their average noise ($p < 2.2$e-16, two-sided Welch's *t* test) and noise plasticity ($p = 6.209$e-05, two-sided Welch's *t* test) are systematically low in comparison with regulated promoters (Fig O in S1 Text).

To test whether all high noise promoters have at least one regulatory input, we calculated what fraction of promoters with noise level over a given cutoff have at least one known

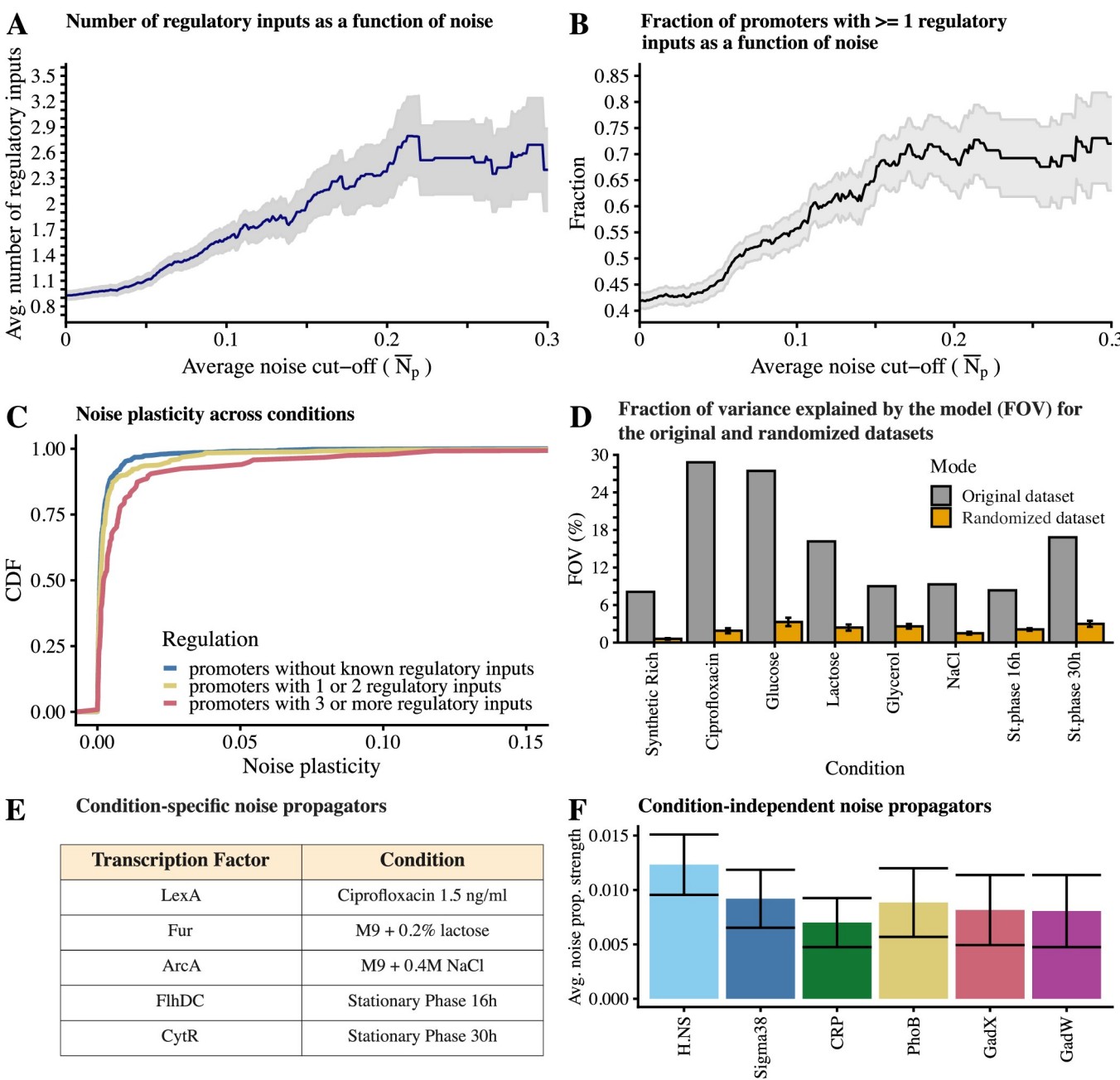

**Fig 4. Noise propagation explains condition-dependent noise levels.** **(A)** More noisy promoters tend to have more regulatory inputs. We sorted promoters by their average noise $\bar{N}_p$ across the 8 conditions and calculated the mean (y-axis) and standard error (gray area) of the average number of TFs known to regulate the promoters with noise level above $\bar{N}_p$, as a function of $\bar{N}_p$ (x-axis). **(B)** The fraction of regulated promoters increases with higher levels of noise. We sorted promoters as in panel A and calculated the fraction (y-axis) and standard error (gray area) of the number of promoters with at least 1 regulatory input with noise level above $\bar{N}_p$, as a function of $\bar{N}_p$ (x-axis). **(C)** The noise plasticity increases with number of regulatory inputs of the promoter. Shown are the cumulative distributions of the variance in noise across the 8 conditions for promoters with no known regulatory inputs (blue), 1 or 2 known regulators (yellow), and 3 or more known regulators (red). **(D)** The Motif Activity Response Analysis model explains a significant fraction of the variation in noise levels. Shown is the percentage of explained variance (FOV %, y-axis) in each of the 8 conditions (x-axis) after running the model on the real dataset (gray bars) and on randomized data (orange bars). Randomized data were generated by shuffling the association between regulatory inputs and expression noise multiple times and shown is the average value obtained +/− its standard error. **(E)** Table of TFs predicted by the model to significantly propagate noise in a condition-specific manner, that is, with $A_{rc} > \delta A_{rc}$ in only one condition. **(F)** Average noise propagation activities ($\bar{A}_r$, y-axis) and their error bars ($\delta \bar{A}_r$, vertical lines) of the strongest 6 noise propagators (with $\bar{A}_r > \delta \bar{A}_r$), sorted by significance ($|\bar{A}_r / \delta \bar{A}_r|$, x-axis), which consistently propagate noise across all 8 conditions. The underlying data for Fig 4 can be found in S1 Data and https://doi.org/10.5281/zenodo.4662163.

regulatory input (Fig 4B and Fig P in S1 Text) and found that 70% to 90% of high noise promoters in each condition have at least one known regulatory input. Given that our current knowledge of the regulatory network only represents one-third of *E. coli*'s TFs, this strongly suggests that most, if not all, of the high noise promoters are indeed regulated.

We next tested to what extent noise plasticity increases with the amount of known regulatory inputs of a promoter. As shown in Fig 4C, we indeed observe that genes with more regulatory inputs show larger noise plasticity compared to genes with few or no known regulatory inputs ($p < 3.7 \times 10^{-10}$, two-sided Welch's *t* test). That is, regulated genes are not only more noisy on average, their noise levels are also more regulated across conditions.

If noise propagation is responsible for the high condition dependence of the relative noise levels across conditions, then it should in principle be possible to explain changes in the relative noise levels of promoters in terms of their regulatory inputs, and changes in the amount of noise that different TFs are propagating in different conditions. We have previously developed a model, called Motif Activity Response Analysis [29,30], which models gene expression in terms of computationally predicted regulatory sites in promoters genome-wide using a simple linear model, to identify which TFs are most important for driving observed gene expression changes across a set of conditions. We here adapted this approach to investigate whether changes in relative noise levels of promoters across conditions can be explained in terms of changes in the "noise propagating activities" of regulators and to identify which TFs are most important for propagating noise in different conditions. In particular, we used the RegulonDB database [28] to set a binary matrix of known regulatory inputs, that is, $S_{pr}$ is 1 when promoter $p$ is known to be regulated by TF $r$ and 0 otherwise. We then model the noise $N_{pc}$ of each promoter $p$ in each condition $c$ as a simple linear function of its known regulatory inputs $S_{pr}$ and the unknown noise propagating activities $A_{rc}$ of each regulator $r$ in each condition $c$:

$$(N_{pc} - \bar{N}_c) = \epsilon + \sum_r (S_{pr} - \bar{S}_r)A_{rc}, \tag{1}$$

where $\bar{N}_c$ is the average noise level of all promoters in condition $c$, $\bar{S}_r$ is the average of $S_{pr}$ across all promoters, and $\epsilon$ is a noise term that is assumed Gaussian distributed with mean 0 and unknown variance. For each condition $c$, we then inferred the noise propagating activities $A_{rc}$ by fitting the model (1) using a Gaussian prior on the activities $A_{rc}$ to avoid overfitting, which allows us to calculate a full posterior probability distribution over the activities $A_{rc}$ [30].

There are many reasons why the crude model (1) is extremely unlikely to provide a good quantitative model for the measured noise levels. First, as already mentioned above, our current knowledge of *E. coli*'s regulatory network is very incomplete with no targets known for almost two-thirds of its TFs, that is, there may well be significantly more regulatory interactions that we do not know about than those that we happen to know about. Second, as discussed in the previous section, the extent to which noise from a given TF propagates to a given target is likely a complex function of the combination of TFs that target a given promoter, the numbers, positions, and affinities of the binding sites for each of these TFs, the concentrations of all these TFs in a given condition, and so on. In particular, it is likely that of all promoters that a given TF targets, only a fraction will be sensitive to the noise in the TF binding in a given condition. However, we currently have no knowledge whatsoever about the extent to which different targets may respond to noise in the TFs that regulate them in a given condition. In absence of such knowledge, Eq (1) makes the crude assumption that each TF will propagate the same amount of noise to all its (known) target promoters and that the total noise of a promoter is simply the sum of the noise propagated by each of the regulators. Note that the latter effectively assumes that the fluctuations in the binding of all TFs are mutually independent, which is also unlikely to be true.

Consequently, the aim of the model (1) is not to explain noise levels of individual promoters or to quantify the amount of noise propagated by each TF. Rather, the aim is to test whether this crude model of noise propagation can explain a significant fraction of the variation in noise levels across promoters and to identify which TFs are most responsible for noise propagation in each condition.

As shown in Fig 4D (gray bars), in spite of our highly incomplete and rudimentary knowledge of *E. coli*'s regulatory network, the simple model explains between 10% and 30% of the variance in noise levels across conditions. To confirm the significance of these results, we fit the same model to data in which the association between regulatory inputs and noise levels were randomized by randomly shuffling the rows of the noise matrix $N_{pc}$ and observed that the fraction of explained variance on the randomized data was always much lower than on the real data (Fig 4D, orange bars).

The model of Eq (1) also calculates error bars $\delta A_{rc}$ for the estimated noise propagation activities $A_{rc}$ of each regulator $r$ in each condition $c$, allowing us to infer which TFs are most significantly propagating noise in each condition and Fig Q in S1 Text shows, for each condition, all TFs for which the noise propagating activity was larger than its error bar, that is, $A_{rc} > \delta A_{rc}$. Note that, while activity $A_{rc}$ corresponds to the average amount of additional noise per target that regulator $r$ is predicted to cause in condition $c$, this should not be interpreted as the typical amount of noise per target. As discussed above, different target promoters will have very different sensitivities to the noise of regulator $r$, so that the $A_{rc}$ reflects an average between weak or no noise propagation at many targets and much stronger noise propagation at a subset of the targets of $r$.

Focusing first on TFs that propagate noise in a highly condition-specific manner, Fig 4E lists the 5 TFs that had significant noise propagating activity in only one condition. For several of these TFs, their known functional role is consistent with the prediction that they propagate noise in these specific conditions. To mention the most obvious case, the TF LexA is predicted to propagate noise only in the sub-MIC ciprofloxacin condition. LexA is a repressor of the SOS response genes, and it is known that ciprofloxacin causes DNA damage and induces the SOS response [31]. Since we employed ciprofloxacin at a concentration well below the minimal inhibitory concentration, DNA damage likely only occurred in a subset of the cells, leading to heterogeneity in LexA activity across the cells. Similarly, the model predicted that FlhDC, the master regulator of flagellar biosynthesis [32], significantly propagates noise only in early stationary phase. It is known that flagellar synthesis peaks toward the end of exponential phase and decreases shortly after entry into stationary phase [33]. Since the 16-h condition is a transition between late exponential growth and entry into stationary phase, it seems plausible that some cells had entered growth arrest and were no longer expressing components of the flagellar machinery, while others had not yet transitioned, causing heterogeneity in the expression of targets of FlhDC. The other examples of condition-specific noise propagators are discussed in the S1 Text.

In addition to condition-specific noise propagators, we noted that many of the most significant noise propagators were found in multiple conditions (Fig Q in S1 Text). To identify regulators that were consistently contributing to noise propagation in all conditions, we calculated, for each regulator $r$, its average noise propagating activity $\bar{A}_r$ averaged over all conditions (SI Methods and Texts in S1 Text). Fig 4F shows the 6 TFs that were most significantly propagating noise in all conditions. As discussed in more detail in the S1 Text, the appearance of many of these TFs likely reflects our experimental setup, that is, growth in minimal media in microtiter plates. For example, the early stationary phase and stress regulator Sigma38 (*rpoS*) has been shown to have heterogeneous activity across single cells in M9 media with glucose [34].

Similarly, limiting oxygen levels in microtiter plates can lead to production of fermentation products [35,36], which are known to acidify the medium [37], explaining the appearance of GadW and GadX, which are involved in the response to acid stress [38]. The prediction that the histone-like TF H.NS is the most significant noise propagating TF is interesting, given that in eukaryotes, noise properties of different genes have been related to nucleosome organization in their promoters [39].

Although the predicted condition-dependent role of these TFs in propagating noise are, at this point, just hypotheses that require in-depth experimental follow-up to confirm, for several cases, the predicted role in noise propagation by these TFs is highly plausible, given their known functional role, and highlights that the simple model can make concrete predictions about which TFs are most involved in driving gene expression noise in different conditions.

In summary, we have presented multiple lines of evidence to confirm that noise propagation plays an important role in determining condition-dependent expression noise genome-wide. Constitutive promoters have consistently low noise and low noise plasticity across conditions. In contrast, across all conditions, we find that the higher the expression noise, the higher the number of known regulatory inputs promoters tend to have. Although almost 60% of promoters have no known regulatory input, 70% to 90% of high noise promoters have at least one known regulatory input. In addition, promoters with more known regulatory inputs also exhibit higher noise plasticity across conditions, indicating that gene regulation causes noise levels to be regulated as well. And finally, in spite of our very limited knowledge of *E. coli*'s regulatory network, a crude model of noise propagation explains 10% to 30% of the variance in relative noise levels across conditions. Together, these results imply that propagation of noise through the regulatory network is a major determinant of condition-dependent expression noise. That is, not only the mean expression levels of genes are determined by gene regulation, the noise levels of genes are to a substantial extent determined by the structure of the gene regulatory network as well.

## Gene features are organized along 2 major axes reflecting average expression and regulation

Previous studies of the genome-wide correlation structure of gene features have uncovered that genes are organized along a one-dimensional axis that relates evolutionary rates, codon bias, and gene expression level [40–43], that is, highly expressed genes tend to have strong codon bias and slowly evolving coding regions, whereas lowly expressed genes tend to have weak codon bias and evolve more rapidly. We next set out to extent such analysis of the genome-wide correlation structure of gene properties, including gene properties associated with gene regulation and expression noise into the analysis, and investigate the interdependence of absolute gene expression, regulation of expression, expression noise, codon bias, and evolutionary rates. We collected a set of features for *E. coli* genes on a genome-wide scale from the literature including the absolute expression levels at both the RNA [6] and protein level [44], sequence properties such as codon bias and the evolutionary rates at both synonymous and nonsynonymous sites (denoted by *dN* and *dS*, respectively) [42], and the number of regulatory inputs of each gene [28]. We then complemented these features with gene expression features that we measured here, including mean expression level, expression plasticity across the 8 growth conditions, the mean expression noise level, and noise plasticity across the 8 growth conditions.

In total, we gathered 10 different gene features and then calculated an overall normalized correlation matrix $R$ of correlations between these features, that is, with $R_{ij}$ the Pearson correlation between features $i$ and $j$. We then performed Principal Component Analysis (PCA) of the

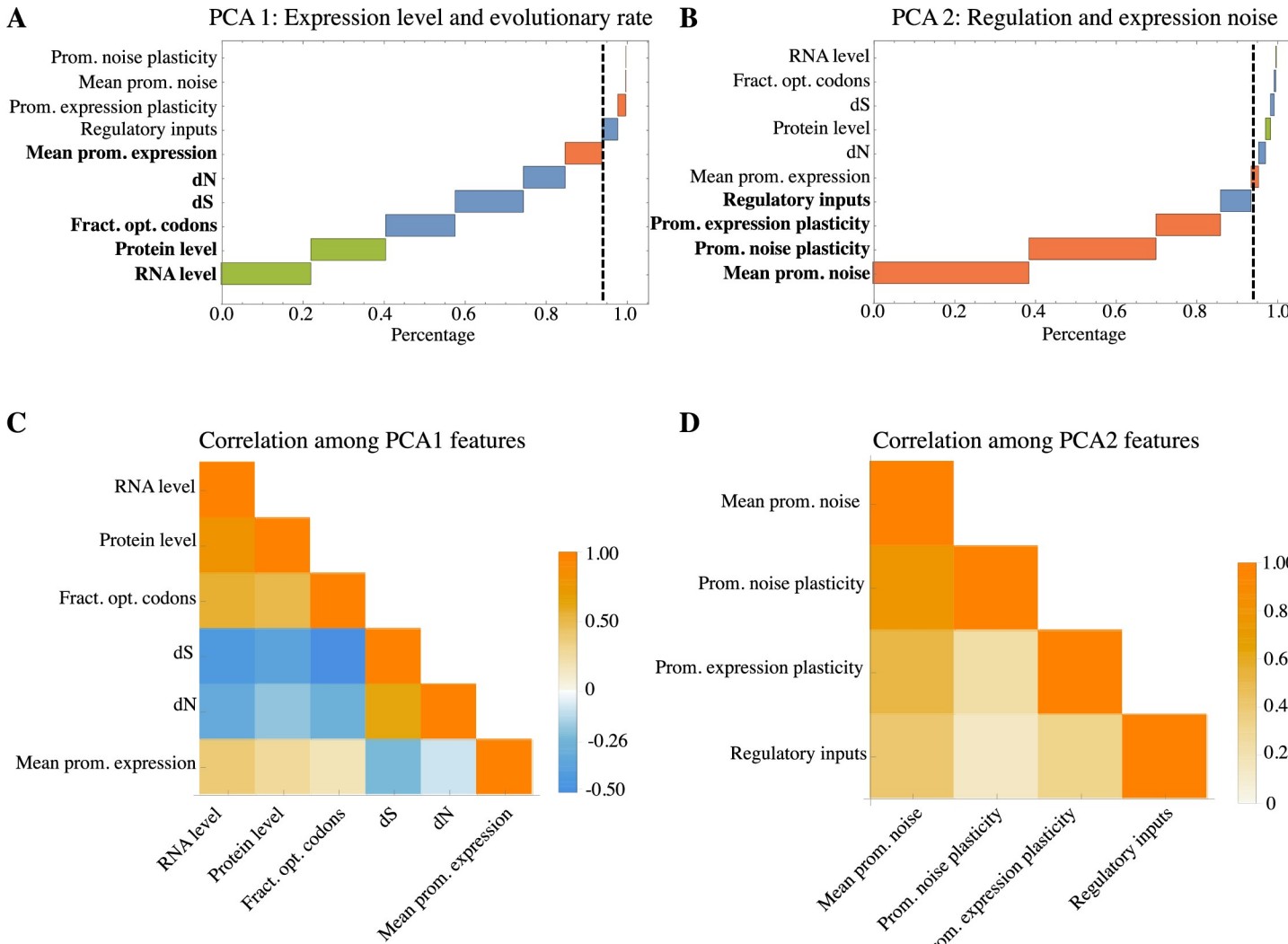

**Fig 5. Principal component analysis shows gene features are distributed along 2 major axes associated with absolute expression level and gene regulation, respectively.** (**A**) Relative contribution of the 10 gene features to the first PCA component, sorted from bottom to top. The features in bold together account for 94% of the first component. In green are expression measurements obtained from previous studies, sequence features are in blue, and features measured in this study are in red. (**B**) As in panel A but now for the second PCA component. (**C**) Correlation structure of the features contributing to the first PCA component. Negative correlations are in blue and positive correlations in orange. (**D**) As in panel C but now for the second PCA component. The underlying data for Fig 5 can be found in S1 Data.

matrix *R* to characterize the overall genome-wide correlation structure of these gene features. As shown in Fig R in S1 Text, the first 2 principal components capture significantly more of the total variance than the other 8 components, capturing more than 50% of the total variance. That is, the plane spanned by these first 2 PCA axes captures the majority of the variation in the 10-dimensional space of gene features. Moreover, each of these axes corresponds to a weighted average of the 10 gene features, and the fact that the axes are (by construction) orthogonal implies that these combinations of gene features vary independently of each other. We thus next investigated which gene features are associated with these first 2 PCA axes. We find that the first PCA axis corresponds precisely to the previously observed organization of genes by their absolute expression levels, codon bias, and evolutionary rates [40–43] (Fig 5A). That is, 94% of the weight along this first PCA component is accounted for by mean RNA and protein levels, codon bias, and evolutionary rates at synonymous and nonsynonymous sites

(Fig 5A). As observed previously, the absolute expression levels and codon bias are positively correlated with each other, while the 2 evolutionary rates *dS* and *dN* are negatively correlated with these features (Fig 5C).

Strikingly, the second PCA axis corresponds almost entirely to gene features associated with gene regulation and expression noise. Around 94% of this vector's weight is accounted for by gene expression noise, noise plasticity, plasticity in mean expression, and number of regulatory inputs (Fig 5B). Moreover, we find that these 4 features are all positively correlated with each other on a genome-wide scale (Fig 5D). That is, this second PCA axis organizes genes by their regulation and expression noise. On one end of this axis are constitutively expressed genes that do not change their mean expression level across conditions and have low noise in all conditions, whereas on the other end of the axis are highly regulated genes that have a high number of regulatory inputs, are highly plastic in expression, and have high and varying expression noise across conditions. This result not only shows that gene regulation and expression noise are intimately coupled on a genome-wide scale, confirming the importance of noise propagation for condition-dependent expression noise, it also shows that these gene regulatory features are varying independently of the absolute expression and evolutionary rate features of the first principal axis.

## Discussion

Although it is now well established that gene expression is an inherently noisy process, so far little is known in bacteria about how noise levels of genes vary across growth conditions. Here, we used high-throughput flow cytometry in combination with a library of fluorescent transcriptional reporters to quantify expression noise of *E. coli* promoters genome-wide. The general picture that emerges from our study is that the expression noise of a given gene in a given condition is the sum of two contributions: a minimal amount of noise that derives from global physiological fluctuations and that is approximately equal for all genes, and a highly gene- and condition-specific component that is substantially due to propagation of noise through the regulatory network. Constitutively expressed genes have least expression noise in each condition and consistently exhibit low noise. Consequently, constitutively expressed genes also exhibit least variation in noise levels across conditions. In contrast, regulated genes exhibit additional noise due to noise propagation. As the regulatory network changes its state across conditions, so does the propagation of noise through the regulatory network, causing regulated genes to change their noise levels in a highly condition-dependent manner. That is, our results suggest that the cell's regulatory network does not only control the mean expression levels of genes across conditions, but also controls the amount of expression noise of each gene, making gene expression noise a regulated quantity. This intimate coupling of expression noise and regulation was underscored by our analysis of the genome-wide correlation structure of various gene features. We found that number of regulatory inputs, expression plasticity, expression noise, and noise plasticity are all positively correlated on a genome-wide scale and that variations in these quantities are indepedent of the correlated variations in average absolute expression, codon bias, and evolutionary rate that has been observed previously [40–43].

We also observed that both the noise floor and the total amount of variation in noise levels systematically decrease with the growth rate of the cells and is highest in the stationary phase (Fig 1). Both its dependence on growth rate, and the fact that this noise floor appears to affect all promoters equally, strongly suggest that the noise floor is driven by global physiological fluctuations, although it is currently not clear which physiological variables contribute most to the noise floor. Our analysis shows fluctuations in cell sizes are similar in all conditions, and our comparison of plasmid-based and chromosomally integrated reporters shows that plasmid

and chromosome copy number fluctuations are either similar in size or do not contribute substantially to the noise floor. However, fluctuations in RNA polymerase concentration, ribosome and charged tRNA concentrations, mRNA decay rates, and fluctuations in growth rate itself are all plausible contributors to the noise floor. In addition, the fact that not only the noise floor but also the total variance in noise levels decreases with growth rate suggests that increased growth may dampen the propagation of noise through the regulatory network. To gain further insight into which fluctuations set the noise floor, and why the noise floor decreases with growth rate, will likely require quantitative time course data, for example, from approaches that combine microfluidics with time-lapse microscopy [45,46].

Although our modeling of noise levels in terms of known regulatory interactions showed that noise propagation can explain a significant fraction of the condition-dependent variation in noise levels genome-wide, there are many questions that remain for future work. Our modeling identified both TFs that appear important noise propagators in all conditions, for example, the histone-like H.NS and sigma factor Sigma38, as well as TFs that significantly propagate noise in one condition only, for example, LexA under treatment with ciprofloxacin and FlhDC in early stationary phase. Therefore, the most obvious direction for detailed experimental follow-up is to investigate the precise role of these TFs in noise propagation. For example, it is currently not clear what the main biophysical mechanism is through which noise is propagated from regulators to their targets. Both fluctuations in TF concentration across cells and the stochastic binding and unbinding of TFs to promoters will contribute to noise propagation, but the relative contribution of these are currently not known. In addition, it is also not clear what sets the sensitivity of different target promoters to fluctuations in an upstream regulator. To quantitatively understand the sensitivities of different target promoters to noise in the activities of their regulators will likely require much more realistic biophysical models of promoter function, which take into account that different TFs compete for binding to the promoter, that binding rates depend on TF concentrations, that interactions between bound TFs and RNA polymerase depend on the relative positioning of sites, and so on. Developing such quantitative models will likely require detailed data on the expression dynamics of different promoter architectures as growth conditions are varied.

Lastly, since the structure of the regulatory network is a major determinant of genome-wide noise levels, this raises the question of how natural selection has acted on noise propagation. One might expect that by making gene regulation less accurate, the effects of noise regulation are mainly deleterious, so that natural selection would be expected to act to minimize noise propagation. However, our previous theoretical work has shown that, by effectively implementing a targeted bet hedging strategy, noise propagation can in fact be beneficial in many circumstances where perfect regulation is difficult to achieve [9]. It is thus conceivable that the way noise propagates through the regulatory network has been tuned by natural selection. It will be interesting to investigate to what extent the condition-dependent noise properties that we have measured contribute to growth and survival of the population in these conditions. For example, it is conceivable that the systematic increase of expression noise as growth rate decreases might be an adaptive strategy by which cells more actively explore different phenotypes when they grow more slowly. Similarly, it would be very interesting to investigate to what extent the noise propagation patterns that we observed in our lab strain of *E. coli* are conserved in related wild bacterial strains or related species.

## Materials and methods

### Strains

All 1,810 strains used in this study were taken from [21] and have been previously described [7]. In short, each strain carries a transcriptional fusion of a given native *E. coli* promoter

followed by a strong ribosomal binding site and *gfp-mut2* on a low copy number plasmid (SI Methods and Texts in S1 Text).

## Growth conditions

The library of strains was grown in a total of 8 different conditions: minimal media, M9 (0.1 mM CaCl$_2$, 1 mM MgS$o_4$, 1 × M9 salts [Sigma M6030]) supplemented with either 0.2% glucose (w/v), 0.2% glycerol (v/v), 0.2% lactose (w/v), 0.4 M NaCl (+0.2% glucose [w/v]), or 1.5 ng/ml ciprofloxacin (+0.2% glucose [w/v]); a MOPS based synthetic rich media (Teknova, M2105) supplemented with 0.2% glucose, and 2 stationary phase conditions, where plates were grown for either 16 h or 30 h in M9 minimal media + 0.2% glucose (w/v) (SI Methods and Texts in S1 Text).

## Flow cytometry quantification of fluorescence

We measured the distribution of GFP fluorescence levels in single cells using a FACSCanto II (BD Biosciences) with a high-throughput sampler (HTS), fluorescence excitation at 488 nm and a 530/30-nm filter for emission. We used a Bayesian procedure that removes outliers to extract the mean and variance of the log-fluorescence distributions as described in [23] (SI Methods and Texts in S1 Text).

## Minimal variance as a function of mean and noise estimation

Flow cytometry data show a clear lower bound on noise levels (variance of log-fluorescence) that depends on the mean of expression. In previous work [9], we derived a functional form for this noise floor as a function of mean expression and used it to correct for the dependency in each condition (see S1 Text). We define a promoter's noise, $N_{pc}$, as the difference between the measured variance and the fitted minimal variance.

## Noise propagation features

We sorted all annotated genes by their average noise across all conditions ($\bar{N}_p$) and as a function of a cutoff in $\bar{N}_p$, we calculated the mean and standard error of the number of regulatory inputs of all genes with $\bar{N}_p$ values above the cutoff and the fraction of genes with at least one known regulatory input. As a measure of noise plasticity of each promoter $p$, we calculated the variance of the noise levels $N_{pc}$ across conditions. We used the same promoter annotation as in [9], where the promoter fragments had been reannotated by mapping the primer pairs used to construct the library to the *E. coli* K12 MG1655 genome (SI Methods and Texts in S1 Text).

## Fitting noise in terms of regulatory inputs

To model noise in terms of regulatory inputs, we adapted a previously developed method, called Motif Activity Response Analysis, which models gene expression levels in terms of computationally predicted regulatory sites in promoters and condition-dependent activities of regulators [29,30]. In particular, we model the noise $N_{pc}$ of each promoter $p$ in each condition $c$ as a linear function of the condition-dependent noise-propagating activities $A_{rc}$ of the regulators known to regulate promoter $p$, that is, Eq (1). Details of the approach are in the SI Methods and Texts in S1 Text.

## Principal component analysis

For each promoter, we gathered a list of 10 features associated with the immediately downstream gene using both the measurements in this study, as well as previously published data. We calculated a covariance matrix containing all the variances of each of the features across genes, and the covariances of each pair of features. We then transformed this covariance matrix into a matrix of correlation coefficients and performed PCA (SI Methods and Texts in S1 Text).

## Supporting information

**S1 Text. Supporting information file containing supplementary methods, supplementary text, and supplementary figures and tables.**
(PDF)

**S1 Data. Excel table containing the underlying data for results presented in the main and supplementary figures.**
(XLSX)

## Acknowledgments

We would like to thank our FACS Core Facility, Guillaume Witz for support with microscopy analysis, and Mikhail Pachkov for help with setting up the environment for MARA. Thanks to Urs Jenal, Gasper Tkačik, and Olin Silander for useful discussions on the project; and Dorde Relic for comments on the manuscript. This work was partly funded by the Werner Siemens Stiftung through a fellowhip to AU, the SystemsX.ch StoNets grant, and the SNF grant 31003A_159673 to EvN.

## Author Contributions

**Conceptualization:** Thomas Julou, Erik van Nimwegen.

**Data curation:** Arantxa Urchueguía.

**Formal analysis:** Arantxa Urchueguía, Luca Galbusera, Dany Chauvin, Thomas Julou, Erik van Nimwegen.

**Funding acquisition:** Erik van Nimwegen.

**Investigation:** Arantxa Urchueguía, Dany Chauvin, Gwendoline Bellement, Thomas Julou, Erik van Nimwegen.

**Methodology:** Arantxa Urchueguía, Luca Galbusera, Dany Chauvin, Thomas Julou, Erik van Nimwegen.

**Resources:** Arantxa Urchueguía, Gwendoline Bellement.

**Supervision:** Thomas Julou, Erik van Nimwegen.

**Validation:** Arantxa Urchueguía.

**Visualization:** Arantxa Urchueguía.

**Writing – original draft:** Arantxa Urchueguía, Thomas Julou, Erik van Nimwegen.

**Writing – review & editing:** Arantxa Urchueguía, Luca Galbusera, Dany Chauvin, Thomas Julou, Erik van Nimwegen.

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
