## [Editor Report · Decision Letter 0]

4 May 2021

Dear Dr van Nimwegen, 

Thank you for submitting your manuscript entitled "Genome-wide gene expression noise in Escherichia coli is condition-dependent and determined by propagation of noise through the regulatory network" for consideration as a Research Article by PLOS Biology. Please accept my apologies for the delay in getting back to you as we consulted with an academic editor about your submission.

Your manuscript has now been evaluated by the PLOS Biology editorial staff as well as by an academic editor with relevant expertise and I am writing to let you know that we would like to send your submission out for external peer review.

Please re-submit your manuscript within two working days, i.e. by May 06 2021 11:59PM.

Kind regards,

Richard

Richard Hodge, PhD

Associate Editor, PLOS Biology

rhodge@plos.org

PLOS

---

## [Decision Letter · Decision Letter 1]

26 May 2021

Dear Dr van Nimwegen,

Thank you very much for submitting your manuscript "Genome-wide gene expression noise in Escherichia coli is condition-dependent and determined by propagation of noise through the regulatory network" for consideration as a Research Article at PLOS Biology. Your manuscript has been evaluated by the PLOS Biology editors, an Academic Editor with relevant expertise, and by three independent reviewers.

The reviews are attached below. You will see that the reviewers find your manuscript interesting and well done, but also raise concerns with the interpretation of the data in several instances and ask that the conclusions are toned down. In addition, Reviewer #1 raises concerns with the strength of advance over your previously published study (Wolf et al, 2015, eLife). We ask that the manuscript is re-written so that the novelty and advance of the current work is made clear.

In light of the reviews, we will not be able to accept the current version of the manuscript, but we would welcome re-submission of a much-revised version that takes into account the reviewers' comments. We cannot make any decision about publication until we have seen the revised manuscript and your response to the reviewers' comments. Your revised manuscript is also likely to be sent for further evaluation by the reviewers.

We expect to receive your revised manuscript within 3 months. 

**IMPORTANT - SUBMITTING YOUR REVISION**

*Re-submission Checklist*

*Published Peer Review*

*PLOS Data Policy*

*Blot and Gel Data Policy*

Sincerely,

Richard

Richard Hodge, PhD

Associate Editor, PLOS Biology

rhodge@plos.org

PLOS

REVIEWS:

Reviewer #1: This is a well-reasoned and well-presented manuscript studying the role of environmental conditions in expression variability in E.coli. The authors hypothesize that a key factor in expression noise is that genes regulated by transcription factors will propagate the noise of the factors and will therefore be more noisy the factors or the number of factors. The ultimate conclusion, that environment affects noise through transcription factors and that it affects different genes differently, is not surprising. 

This manuscript builds heavily upon this group's previous work. The manuscript is well written, and the clarity of data analysis and presentation of supporting evidence is exemplary. Overall, the manuscript is impressive in my opinion has the potential to support many interesting developments, both theoretical and experimental. I do have, however, several important concerns that should affect the editorial decision. Once these major issues are addressed, I'll be happy to review the manuscript again.

1. Some of the results presented are quite similar to the senior author's older work "Expression noise facilitates the evolution of gene regulation" (Wolf et al., eLife 2015). Indeed, the following quote is from that eLife 2015 publication digest entry: "Wolf et al. also found that noisy promoters tend to be highly regulated by transcription factors: the proteins that control gene expression by binding to promoter regions." This sounds much like the take-home message of this manuscript, which raises questions regarding the contribution in this manuscript. Unfortunately, the authors have made my life very difficult as a reviewer, by blending old and new methods in their manuscript in a way that makes it difficult to assess the novelty in this work. For example: most of Fig. 1 in this manuscript is very similar to Fig. 1 of Wolf. Fig. 2B in this manuscript is very similar to Fig. 1E in Wolf. Fig. 4A in this manuscript is very similar to Fig. 2A in Wolf. Fig. 4B in this manuscript is very similar to Fig. 2C in Wolf. There are more examples. The fact that as deep as Fig. 4 in this manuscript I still see analyses that are very similar to this group's work from 6 years ago, is worrying. It suggests to me that the authors need to rewrite the manuscript in such a way that makes it utterly clear what the novelty in this work. This would be a major edit since it is expected that non-novel methods/analyses belong either in the introduction or in the supplementary material, and that the results section contains only the novel results for this work. Only after the authors have clearly separated the old from the new, will I be able to properly judge the significance of the new manuscript. As it stands currently, my impression is that the major breakthrough was in the 2015 paper and that this followup work simply implements those methods in more experimental conditions.

2. I think the manuscript can be made more robust by softening some of the interpretations the authors make about the data. The data as they are speak eloquently and I fear these particular interpretations may detract and even may misguide and so should be removed.

* Fig 1C - there not enough data to justify a linear fit. These data could be explained by a number of plausible models, for example a Hill function.

Similarly, the interpretation in the text is not necessary. Unless the authors suggest a plausible linear relation, and then extract the slope of the linear fit and discuss its implications (adjusting for the correct units and considering fit confidence levels) then I suggest the authors do away with this fit altogether. 

* Since Growth rate ~ size and noise floor ~ 1/sqrt(size), doesn't that imply that the faster the growth the lower the noise? (e.g. Fig. 1C). This seems like an obvious statement, and so should be made clear. Again it questions the validity of the linear fit in Fig. 1C.

* Fig 2E-F - I'm not sure the evidence supports the statements - "Noise/Expression plasticity". How different are they from A-B? One outlier is hardly convincing. Do the authors really want to go into the statistical analysis of whether they are or aren't different from the A-B case? Please remove/adjust or otherwise support the statements with a rigorous statistical analysis. 

Reviewer #2: Review: In the manuscript Urchueguía et al study gene expression noise in E. coli using a plasmid-based reporter system and flow cytometry. Using their system, the authors recapitulated what is known about noisy gene expression (variation among genes, increase of noise with lower expression levels, and a minimal noise floor) and expanded it to multiple growth conditions (faster growth reduces the overall noise level). While the choice of methods is tricky, both plasmids copy number is inherently noisy, and so is flow cytometry, the analysis appears well done.

Fig. 2 is a nice illustration of the behavior of different promoters and Fig. 3 states the authors hypothesis that noise propagation increases noise clearly. The analysis of the effect of regulation on noise (Fig 4) yielded interesting results. Noisy genes generally had more regulatory input, and highly regulated promoters showed higher noise plasticity. 

The authors interpret their findings in Fig. 4 as 'Noise propagation explains condition-dependent noise levels', which at this stage I cannot see that the authors have shown. In Fig. 4C the authors show the fraction of explained variance of their simple noise propagation model, which captures around 10 to 30% of the observed noise. With 90-70% of the observed noise unaccounted for, it is hard to argue that noise propagation has explained variations in noise levels. I believe that is OK for the authors to only explain noise levels partially, and I also believe that Fig. 4A&B are interesting new data. 

Fig. 4D-E are accompanied by a large section in the text (p. 10) explaining which transcription factor is important and noisy for certain reasons in each condition. While some of the claims could be true (flagellar master regulator being turned on at end of growth -> noisy in early stationary phase), most others are not trivial (ArcA upregulated at high salt, because cell turns fermentative. But why the noise?). This section is very speculative, and could leave the reader more confused than enlightened. I recommend removing it from the main text.

The authors should make clearer what the analysis of regulators (transcription factors and sigma factors) found means for the noise levels. In Fig. 4E the average noise propagation strength for a 'hit' is 0.01, but it was not clear to me if this is little or a lot. After going through the manuscript again I now understood that a gene regulated by this TF/sigma increases its noise level by approx. 0.01 (Eq. 1). Because the typical scale is 0-0.15 in noise level for a gene (Fig. 1D), regulation increases noise by around 10%. A proper estimation of what the numbers mean could benefit the reader.

Finally, Fig. 5 seems ok, but very confusing and needs more explanation. What are dS, dN? I assume codon bias of synonymous and non-synonymous sites, but it is not clear to me what the metric is. What does it mean that genes cluster along the two principal components? Is this trivial, because several noise metrics were included in a data set where the other metrics are highly correlated (protein, RNA expression, dS, dN), or is it non-trivial? Is the conclusion that noise is independent of expression level? What does it mean that regulatory input is only a small component of PCA2?

Overall, the manuscript is fine and can be published after revisions. In addition to the above comments, I want to express a concern that that the claims of the authors, especially in title and abstract, are not fully supported by the data.

Expression noise, as far as I can see, is only 'partially determined' by noise propagation (10 to 30%, if I understand the paper right), so the title should be adapted to reflect that. The abstract should include that instead of a 'significant fraction' of the variation in noise levels, about 10-30% can be explained by the authors. Being transparent on the findings is important, because it will leave other authors (or the same) the opportunity to continue to work on what determines the remaining 70-90% of the noise.

Furthermore, I want to authors to state in the abstract that their system is plasmid-based, because plasmid themselves are highly noisy, and the reader needs to be aware of this limitation of the study. In the main text, the authors should include a discussion of plasmid copy number variations and compare the noise from plasmid copy numbers to their observed noise (e.g. see https://www.nature.com/articles/s41467-021-21734-y). A discussion of the impact of cell size, would also benefit the reader. In exponential growth, cells post- and pre-division vary by a factor of ~2, which might impact the noise recorded by the authors. The impact of the flow cytometer on noise levels would also be important for the readers to judge noise levels.

The statement: "Whereas constitutive promoters exhibit noise near this `noise floor' in all conditions […]" confused me. Does it refer to Fig. 4A, which shows that less noisy promoters are regulated on average by fewer inputs? If so, then the authors have only shown that promoters with lower noise are usually constitutive promoters. Replotting the existing data as number of regulators vs. noise, and then specifically discussing constitutive promoters should be sufficient to test if indeed unregulated promotors are near the noise-floor for in all conditions. Similarly, the statement "regulated promoters are systematically more noisy" should be changed to 'noisier promoters are regulated by an average higher amount of inputs', or alternatively, the can replot their data. 

In addition, a correlation plot, ideally as part of Fig. 4, between noise/noise plasticity and number of regulatory inputs would greatly benefit the readers. A major issue with the claims of the authors is that it is not clear whether the claims are statistically significant. Inclusion of statistical significance tests concerning correlations between noise and regulatory inputs, should be included in the revision. 

The statement: "Our results show that expression noise levels are themselves regulated and determined by the propagation of noise through the gene regulatory network", is, as far as I can see, not shown in the manuscript. Same claim in the discussion: "[…] gene expression noise is […] a regulated quantity". If the authors want to make this statement, then they should include a direct test of this hypothesis, e.g. by measuring the noise level in synthetic constructs with a varying number of regulatory inputs, or any other experimental verification. 

There are several statements in the manuscript that claim that condition-dependent gene expression noise is explained by propagation of noise through the regulatory network, e.g.

"First, all promoters exhibit a condition-dependent minimal amount of noise, and

unregulated promoters typically exhibit noise near this noise floor. Regulated promoters exhibit additional noise that results from propagation of noise through the regulatory network, causing different promoters to show condition-dependent noise levels depending on their regulatory inputs. These results show that not only the mean expression levels of genes are determined by gene regulation, the noise levels of genes are also to a large extent determined by the structure of the gene regulatory network as well." p. 11

The authors should be careful with this claim. If the authors want to make this statement, then they need to show that 1) unregulated promoters are at the noise-floor. 2) only regulated promoters are above the noise floor. I presume that this is based on Fig. 4, but I cannot see which data supports this claim. It could well be that the existing analysis proves the authors point and that I am missing something. Either way, the logic needs to be clearly spelled out and comprehensible.

Finally, it is not clear to me how the authors claim that 'that condition-dependent gene expression noise is explained by propagation of noise through'. This is the major claim of the paper, and is found throughout the paper, including title and abstract. What is the evidence that noise propagation is responsible for the condition-dependent expression noise?

Reviewer #3: This work is a tour de force in characterizing the mean and the variability of E. coli gene expression across many promoters and many experimental conditions. Bravo! This will be a very useful resource for researchers in the field, especially since all of the data have been uploaded by the authors. I am not an experimentalist, and hence cannot assess the experimental parts of the work, which are its cornerstone. The modeling/analysis is rather straightforward, based on previous work from the same group and from other authors, and was easy to follow. I don't have any specific comments there. Based on my ability to understand the computational analysis, and on the beauty of the dataset, I think the manuscript should be accepted to the journal with minimal changes.

1. As mentioned above, I've been able to follow most of the analysis. However, one argument made me pause. Specifically, the authors say that "Noise propagation predicts that relative noise levels are condition-dependent", so that "genes that are regulated by more TFs are expected to generally exhibit more expression noise than genes that are regulated by fewer TFs". This contradicts the model and the experiments on eukaryotic systems, such as in https://elifesciences.org/articles/59351, which argues that multiple enhancers decrease the promoter variability. There are certainly many differences between eukaryotes and prokaryotes -- and it would be worthwhile, I think, for the authors to explain why different results are expected for them under seemingly very similar conditions.

2. Towards the end of the manuscript, I got overwhelmed by the data, and was looking for a theoretical framework to incorporate them into, to help internalize them. Why is the organization of regulation as discovered her? Alas, the Discussion section was very short, and, to a large extent, it simply restated the findings of the Results section, without speculations about the framework to understand the data. It might be that the authors do not have a theoretical framework in mind, which I would understand. However, if they do have some ideas, I would strongly urge the authors to put them on paper, to help us summarize the data within some explanatory framework.

---

## [Decision Letter · Decision Letter 2]

4 Nov 2021

Dear Erik,

Thank you for submitting your revised Research Article entitled "Genome-wide gene expression noise in Escherichia coli is condition-dependent and determined by propagation of noise through the regulatory network" for publication in PLOS Biology. I'm handling this paper temporarily while my colleague Dr Richard Hodge is out of the office. We have now obtained advice from the original reviewers and have discussed their comments with the Academic Editor. 

Based on the reviews, we will probably accept this manuscript for publication, provided you satisfactorily address the remaining points raised by the reviewers. Please also make sure to address the following data and other policy-related requests.

IMPORTANT: Please attend to the following:

a) Please address the remaining request from reviewer #1.

b) Please supply a blurb according to the instructions on the submission form.

c) Please address my Data Policy requests below; specifically, please supply numerical values underlying Figs 1BCDE, 2ABCDEFG, 4ABCDF, 5ABCD, S1AB, S2AB, S3, S4AB, S5A, S6A, S7AB, S8, S9AB, S10AB, S11ABCD,S12, S13AB, S14, S15ABC, S16, S17, S18, and cite the location of the data clearly in each relevant Fig legend (I note that the raw data and code are already provided, but we’ll need the above output values too).

We expect to receive your revised manuscript within two weeks. 

*Published Peer Review History*

*Early Version*

Best wishes,

Roli

Roland Roberts PhD

Senior Editor

PLOS Biology

rroberts@plos.org

on behalf of

Richard Hodge,

Associate Editor,

rhodge@plos.org,

PLOS Biology

DATA POLICY:

We note that raw data and code are presented in the Zenodo deposition. However, we also need the numerical values presented in the figures to be made available in one of the following forms:

Regardless of the method selected, please ensure that you provide the individual numerical values that underlie the summary data displayed in the following figure panels as they are essential for readers to assess your analysis and to reproduce it: Figs 1BCDE, 2ABCDEFG, 4ABCDF, 5ABCD, S1AB, S2AB, S3, S4AB, S5A, S6A, S7AB, S8, S9AB, S10AB, S11ABCD,S12, S13AB, S14, S15ABC, S16, S17, S18 NOTE: the numerical data provided should include all replicates AND the way in which the plotted mean and errors were derived (it should not present only the mean/average values).

We require the original, uncropped and minimally adjusted images supporting all blot and gel results reported in an article's figures or Supporting Information files. We will require these files before a manuscript can be accepted so please prepare and upload them now. Please carefully read our guidelines for how to prepare and upload this data: https://journals.plos.org/plosbiology/s/figures#loc-blot-and-gel-reporting-requirements 

DATA NOT SHOWN?

REVIEWERS' COMMENTS:

Reviewer #1:

The authors have addressed my concerns well. 

I appreciate the edits made and personally I find the manuscript much more readable now, and the contribution in this work clearer. The manuscript remains impressive with exemplary analysis practices. I am happy to recommend publication but I do have one comment - I do not need to see the manuscript again - I leave its final form up to the authors and editor.

The authors insist on fitting a straight line to Fig.1C despite acknowledging that they have no theory for it. They then ask me (in their rebuttal) why would I suggest a more complicated form instead. The answer is, that the straight line would go negative for a fast enough growth rate, and simply saying that it's linear within the relevant range for E.coli is hiding that fact under the rug. Together with very sparse data, lacking a reasoned model, such a fit is damaging to the field. This is a good paper in a respected journal - imagine the number of people who would implicitly assume a linear dependence here with little to no justification. Once such a view becomes "lore" it is very difficult to uproot. I request that they remove the ill-argued and quite possibly misleading linear fit from Fig1C. It is not necessary for the main messages of the paper. It is quite enough to state "decreasing" without stating a precise relation. You may even state "approximately linear" or "appears linear" without actually putting the fit with confidence intervals on the figure. 

Reviewer #2:

The authors have put considerable effort into improving the manuscript. The revised manuscript is a great contribution to the field and can be published without further changes.

Reviewer #3:

My comments on the previous version were pretty minor to start with, and they have been addressed now. I recommend publication.

---

## [Editor Report · Decision Letter 3]

23 Nov 2021

Dear Erik,

On behalf of my colleagues and the Academic Editor, Nathalie Balaban, I am pleased to say that we can in principle accept your Research Article "Genome-wide gene expression noise in Escherichia coli is condition-dependent and determined by propagation of noise through the regulatory network" for publication in PLOS Biology, provided you address any remaining formatting and reporting issues. These will be detailed in an email that will follow this letter and that you will usually receive within 2-3 business days, during which time no action is required from you. Please note that we will not be able to formally accept your manuscript and schedule it for publication until you have any requested changes.

PRESS

Best wishes,

Richard 

Richard Hodge, PhD

Associate Editor, PLOS Biology

rhodge@plos.org

PLOS
